# Cross-sectional associations of physical frailty with fall, multiple falls and fall-injury among older Indian adults: Findings from LASI, 2018

**Shriya Thakkar[1], Muhammad T. [ID][2]\*, Shobhit Srivastava[ID][2]**

**1** Louisiana State University, Baton Rouge, Louisiana, United States of America, **2** International Institute for Population Sciences, Mumbai, Maharashtra, India

\* muhammad.iips@gmail.com, muhammadvallit@gmail.com

## Abstract

### Background

Although there has been a range of studies that focused on physical frailty and associated fall outcomes within developed countries, similar studies from developing countries have been limited. This study aimed to examine the relationship between physical frailty and the prevalence of falls, multiple falls and fall-related injuries among the ageing population within the Indian context.

### Methods

Individual-level data from the first wave of the Longitudinal Aging Study in India (LASI) with 28,285 older adults aged 60 years and above (male 48.9%) was used for this study. Physical frailty was assessed through the physical frailty phenotype adapted from Fried's criteria. Multivariable logistic regression was employed to examine the association of frailty status with falls, multiple falls, and fall-related injuries among Indian older adults.

### Results

The prevalence of frailty was found to be 29.94% within the sample and frail older adults had a higher prevalence of falls (15.43% vs 11.85%), multiple falls (7.73% vs 5.25%), and fall related injuries (6.68% vs 5.29%). The odds of falling among frail older adults were significantly higher in reference to the odds of falling among non-frail older adults [aOR: 1.24; CI: 1.09–1.41]. Similarly, the odds of multiple falls among frail older adults were significantly higher in reference to the odds of multiple falls among non-frail older adults [aOR: 1.24; CI: 1.05–1.48]. Moreover, the odds of fall-related injury among frail older adults were significantly higher in reference to the odds of fall-related injury among non-frail older adults [aOR: 1.21; CI: 1.01–1.45]. Falls, multiple falls and fall-related injuries were found to be significantly associated with employment and poor self-rated health, whereas, females and lone living older adults had a significantly higher likelihood of suffering from falls and multiple falls.

**Data Availability Statement:** The study uses secondary data from the LASI survey collected by the International Institute for Population Sciences (IIPS), Mumbai, which is available in the IIPS LASI

data repository and accessible on reasonable request through https://www.iipsindia.ac.in/content/lasi-wave-i.

**Funding:** The author(s) received no specific funding for this work.

**Competing interests:** The authors have declared that no competing interests exist.

**Abbreviations: aOR**, Adjusted odds ratio; **CI**, Confidence interval; **MPCE**, Monthly per capita consumption expenditure; **SRH**, Self-rated health.

## Conclusion

Older individuals with physical frailty were found to be at increased risk of falls, multiple falls and fall-related injuries in India. The findings of our study also have important clinical implications in the measures undertaken to reduce falls and enable future healthcare practitioners and policymakers to factor in the key determinant of physical frailty.

## Background

The rapid acceleration of population ageing worldwide has created new forms of health-related challenges both in developed and developing nations. The ageing population is at increased risk of age-related physiological degeneration along with other chronic health diseases such as atherosclerosis and cardiovascular disease, arthritis, hypertension, cataracts, osteoporosis, cancer and Alzheimer's disease. In addition, the clinical condition of physical frailty has emerged as the most complex form of expression associated with population ageing. Physical frailty is defined as the state of increasing vulnerability that develops predominantly among the ageing population as a consequence of age-related decline in multiple physiological systems [1]. Lately, the notion of frailty has been part of multiple discourses owing to its multidimensionality as a concept which points to its bio-psychosocial dynamic nature ranging from life-course disease(s) to frailty, and ultimately, the related adversities [2, 3]. With increased life expectancy owing to the recent medical advancements in the domain of public health, there is a projected increase of aged adults (>60 years) from 605 million to 2 billion between 2000 and 2050 globally [4]. Further, there is an expected quadruple increase of individuals over the age of 80 years during the similar timeline [5].

Although the popular assumption holds that frailty is positively correlated with age, however, prior research predicts that frailty is more popularly observed among women than men, and prevalent within existing chronic conditions [6, 7]. According to a study based on the nationally representative data in the United States, it was found that fifteen percent of the ageing population is frail, and there was a steep age-related positive increase of frail individuals from 9% between ages 65–69 to 38% among individuals aged 90 or older [8]. In a systematic review of studies of community-based cohorts, it was found that the prevalence of frailty among community-dwelling older adults ranged from 4% to 59.1%, mainly due to differences in operationalization of frailty status [9]. Further, the impact of frailty on older adults can often have adverse health outcomes accompanied with falls [10], dementia [11], and disability [12, 13]. However, falls tend to be the ruling cause of mortality among the ageing individuals among men and women alike [14]. Further, individuals who have fallen once are often at a risk of falling again [15]. And such falls and fear of falling not only impairs the quality of life, but also contributes to increased expenditure on health care costs and hospitalization [16, 17].

Physical frailty has been popularly examined through the two most widely used measures, namely, the frailty phenotype developed by Fried et al. (2001) utilizing the Cardiovascular Health Study data [6], and the frailty index (FI), developed by Mitnitski et al. (2001), in their Canadian Study of Health and Aging (CSHA) [18]. While the frailty phenotype describes frail individuals to be suffering from some of the reported physical conditions such as, weakness, exhaustion, low physical activity, slow walking speed, and unintentional weight loss [6]; FI assumes that frailty is a condition caused by the increase in health adversities accumulated by an individual over the course of his/her life [18]. Thus, an individual is likely to be frail if he/she is suffering from more negative health outcomes.

According to the previous studies, ageing population above 60 years in India are highly prone to risks of falls which varied from 14% to 53% [19, 20]. Further, fall cases are most likely to be under-reported and it has been found that the number of actual falls is presumably higher than the ones documented [21]. With fall related injuries being highly underreported, there has been an increasing concern for lack of fall related evidence among older adults in developing nations like India, which further does not account for any standardized terminology and practices [22]. Documentation of fall injuries remain insufficient, owing to the lack of valid generalizable evidence, and thus, fall prevention has not been a guiding priority within health policy framework in many developing nations [23].

In this study, we aimed to examine the prevalence of fall, multiple falls and fall-related injuries among the ageing population in India. We also explored the association of physical frailty with fall outcomes among older Indian adults. We hypothesized that physical frailty is significantly positively associated with reporting fall, multiple falls and fall-injury among older adults in India. The conceptual framework based on the previous literature has been developed and summarized in Fig 1.

## Methods

### Study participants

The present study utilizes the individual-level data from the first wave of the Longitudinal Aging Study in India (LASI) conducted during 2017–18. The LASI is a country-representative longitudinal survey of more than 72000 adults aged 45 years and over across all states and union territories of the country that provides vital information on the social, physical, psychological, and cognitive health of the Indian aging population. The LASI survey was conducted

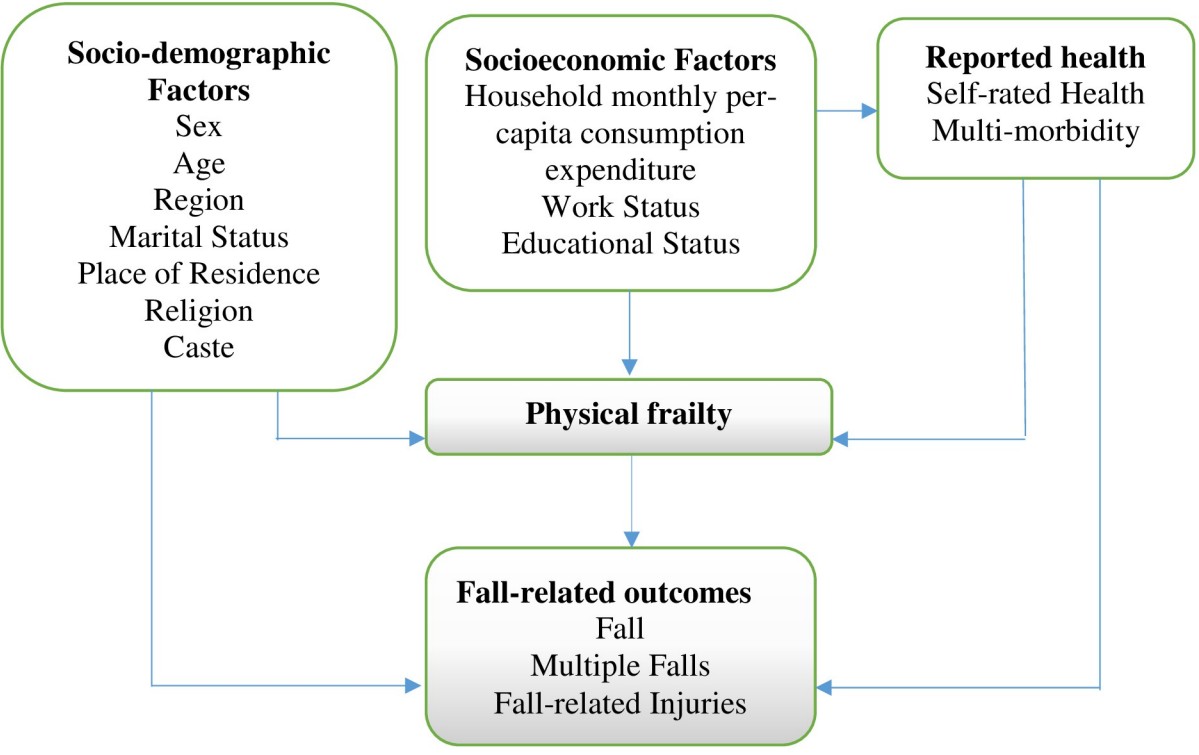

**Fig 1. Conceptual framework of the study.**

through a partnership of the International Institute for Population Sciences (IIPS), Harvard T. H. Chan School of Public Health (HSPH), and the University of Southern California (USC). In the LASI wave 1, the sample selection is based on a multistage stratified cluster sample design, including a three-stage sampling design in rural areas and a four-stage sampling design in urban areas. The details of sampling design, survey instruments, and data collection procedures are provided elsewhere. The present study is conducted on the eligible respondents aged 60 years and above. Thus, the total sample size for the present study after dropping the missing observations of outcome variables was 28,285 (13,836 males and 14,449 females) older adults aged 60 years and above.

## Measures

**Outcome variable.** Falls among older adults in the last two years were self-reported. It was assessed through the question, 'In the past two years, have you fallen down?' The responses were coded as 'no' and 'yes'. Further, information on multiple falls was captured from the following question on number of falls in the last two years. Those reported greater than one fall were categorized as having multiple falls. Additionally, fall-related injuries were assessed by the following survey question that 'In that fall, did you injure yourself seriously enough to need medical treatment? The responses were again coded as 'no' and 'yes'.

**Main exposure variable.** Physical frailty among older adults was the main exposure variable in the study. It was assessed using the physical frailty phenotype which was adapted from the phenotype developed by Fried and colleagues. Accordingly, a composite physical frailty score was generated by constituting five major components; i) self-rated exhaustion, ii) unintentional weight loss, iii) weak grip strength, iv) self-reported low physical activity, and v) slow walking speed. The total frailty score lies between 0 and 5. We classified individuals as "frail" if the score was three or higher and as "non-frail", otherwise. Smedley's handgrip dynamometer was used to measure handgrip strength, and the average score of two successive trials of the dominant hand was considered after adjusting for gender and Body Mass Index (BMI). A 4-meter walk test was conducted twice for measuring the walking speed, and the average seconds to complete the walk was considered after adjusting for gender and height. All the other components of the physical frailty were based on self-reported responses.

**Other covariates.** Socio-demographic variables included age (recoded as 60–69, 70–79 and 80+), sex (male and female), education (recoded as no education, primary education and secondary/higher education), marital status (recoded as married, widowed and others which included separated, divorced and never married), living arrangements (recoded as living alone, with spouse, with spouse and children and others) and work status (recoded as never worked, currently working, not working and retired). Self-rated health (SRH) was available in 5-item scale which are; very good, good, fair, poor and very poor. The variable multi-morbidity was created referring to the presence of two or more of those available chronic diseases which include hypertension, chronic heart diseases, stroke, any chronic lung disease, diabetes, cancer or malignant tumor, any bone/joint disease or any neurological/psychiatric disease. The diseases were self-reported as was assessed through the question "Has any health professional ever diagnosed you with the following chronic conditions or diseases?" And such patient-reported health outcomes are often considered as of high quality and more sensitive than several clinical outcome measures.

Further, the monthly per capita consumption expenditure (MPCE) quintile was assessed using household consumption data. Sets of 11 and 29 questions on the expenditures on food and non-food items, respectively, were used to canvas the sample households. Food expenditure was collected based on a reference period of seven days, and non-food expenditure was

collected based on reference periods of 30 days and 365 days. Food and non-food expenditures have been standardized to the 30-day reference period. The MPCE is computed and used as the summary measure of consumption. The variable was then divided into five quintiles i.e., from poorest to richest. Religion was coded as Hindu, Muslim, Christian, and Others. Caste was recoded as Scheduled Caste/Scheduled Tribe (SC/ST), Other Backward Classes (OBC), and others. The SC refers to the population that is socially segregated and financially/economically weak by their low status as per the caste hierarchy. Similarly, the ST refers to the indigenous populations who are considered among the most disadvantaged and discriminated socio-economic groups in the country. The OBC is the group of people who are identified as 'socioeconomically and educationally backwards'. The other caste category is identified as having higher social status, mostly belong to upper caste categories. Place of residence was coded as urban and rural. Also, the regions of the country were coded as North, Central, East, Northeast, West, and South.

### Statistical analysis

Descriptive statistics along with results of cross-tabulations were presented in the study. We used Chi-square tests to compare the intergroup differences and report the significance level [24, 25]. Additionally, three separate multivariable logistic regression analyses were performed to find out the independent associations of the outcome variables (fall, multiple falls and fall-injury) with physical frailty. These regression models were adjusted for age, sex, marital status, living arrangements, education, work status, SRH, multi-morbidity, household MPCE quintiles, religion, caste, residential status and regions. The binary logistic regression model is usually presented as follows:

$$\text{Logit}\left[P(Y=1)\right] = \beta_0 + \beta * X$$

The parameter $\beta_0$ estimates the log odds of fall, multiple falls and fall-injury for the reference group, while $\beta$ estimates the maximum likelihood, the differential log odds of the outcomes variables associated with a set of predictors X, as compared to the reference group [26]. The estimates were presented in the form of adjusted odds ratio (AOR) with 95% confidence interval (CI). The statistical analysis was performed using Stata 15.1. The individual weights were used for computing the estimates to make them nationally representative.

### Results

Table 1 represents the sample distribution of the study population. About 30% of the older adults were physically frail in the current study. About 5.1% of the older adults were living alone. Almost 8.9% of the older adults were retired and 28% never worked. About 3.1% and 21% of the older adults reported very poor and poor self-rated health respectively. About 24% of the older adults suffered from multi-morbidity.

Table 2 represents the prevalence of fall, multiple falls and fall-injury by background characteristics among older adults. The overall prevalence of fall, multiple falls and fall-related injuries was 13%, 5.6% and 5.8% respectively. The prevalence of fall (15%), multiple falls (7.7%) and fall related injuries (6.7%) was higher among older adults who were frail.

Table 3 represents the multivariable logistic regression estimates of fall, multiple falls and fall-injury by socioeconomic and health characteristics among older adults. The odds of falling among frail older adults were significantly higher in reference to the odds of falling among non-frail older adults [aOR: 1.24; CI: 1.09–1.41]. Similarly, the odds of multiple falls among frail older adults were significantly higher in reference to the odds of multiple falls among non-frail older adults [aOR: 1.24; CI: 1.05–1.48]. Moreover, the odds of fall-related injury

**Table 1. Sample distribution by background characteristics.**

| Variables | Distribution | |
|---|---|---|
| | **Frequency** | **w%** |
| **Frailty status*** | | |
| No | 19,781 | 70.06 |
| Yes | 7,782 | 29.94 |
| **Age (in years)** | | |
| 60–69 | 17,179 | 60.74 |
| 70–79 | 8,172 | 28.89 |
| 80+ | 2,934 | 10.37 |
| **Sex** | | |
| Male | 13,836 | 48.92 |
| Female | 14,449 | 51.08 |
| **Marital status** | | |
| Currently in union | 18,110 | 64.03 |
| Widowed | 9,404 | 33.25 |
| Others | 771 | 2.73 |
| **Living arrangement** | | |
| Alone | 1,431 | 5.06 |
| With spouse | 5,627 | 19.89 |
| With spouse and children | 12,259 | 43.34 |
| Others | 8,968 | 31.71 |
| **Educational status** | | |
| None | 15,036 | 53.16 |
| Primary | 5,218 | 18.45 |
| Secondary/higher | 8,031 | 28.39 |
| **Work status** | | |
| Never worked | 7,777 | 27.50 |
| Not working | 9,769 | 34.54 |
| Working | 8,225 | 29.08 |
| Retired | 2,514 | 8.89 |
| **Self-rated health*** | | |
| Very good | 977 | 3.72 |
| Good | 8,694 | 27.67 |
| Fair | 12,082 | 44.73 |
| Poor | 5,298 | 20.83 |
| Very poor | 732 | 3.06 |
| **Multi-morbid*** | | |
| No | 21,435 | 76.36 |
| Yes | 6,837 | 23.64 |
| **MPCE quintile** | | |
| Poorest | 5,876 | 20.77 |
| Poorer | 5,834 | 20.63 |
| Middle | 5,746 | 20.31 |
| Richer | 5,524 | 19.53 |
| Richest | 5,305 | 18.76 |
| **Caste** | | |
| SC/ST | 9,371 | 33.13 |
| OBC | 10,621 | 37.55 |

(*Continued*)

**Table 1.** (Continued)

| Variables | Distribution | |
|---|---|---|
| | Frequency | w% |
| Others | 8,293 | 29.32 |
| **Religion** | | |
| Hindu | 20,567 | 72.71 |
| Muslim | 3,350 | 11.84 |
| Others | 4,368 | 15.44 |
| **Place of residence** | | |
| Urban | 9,781 | 34.58 |
| Rural | 18,504 | 65.42 |
| **Region** | | |
| North | 5,318 | 18.80 |
| Central | 3,829 | 13.54 |
| East | 4,860 | 17.18 |
| Northeast | 3,469 | 12.26 |
| South | 6,905 | 24.41 |
| West | 3,904 | 13.80 |
| **Total** | 28,285 | 100 |

Notes:

*sample size may differ due to missing cases;

%: Percentage; w%: weighted percentage prevalence to account for survey design and provide national population estimates; MPCE: Monthly per capita consumption expenditure

among frail older adults were significantly higher in reference to the odds of fall-related injury among non-frail older adults [aOR: 1.21; CI: 1.01–1.45]. The odds of falling and multiple falls among older females were significantly higher in reference to the odds of falling [aOR: 1.23; CI: 1.07–1.41] and multiple falls [aOR: 1.37; CI: 1.13–1.67] among older male. The odds of falling and multiple falls among older adults living with spouse and children were significantly low in reference to the odds of falling [aOR: 0.47; CI: 0.26–0.85] and multiple falls [aOR: 0.41; CI: 10.18–0.92] among older adults living alone. The odds of fall, multiple falls and fall-injury among older adults who were currently working were significantly higher in reference to the odds of fall [aOR: 1.55; CI: 1.31–1.85], multiple falls [aOR: 1.62; CI: 1.28–2.04] and fall-injury [aOR: 1.67; CI: 1.30–2.13] among older adults who never worked. The odds of fall, multiple falls and fall-injury among older adults who reported very poor SRH were significantly higher in reference to the odds of fall [aOR: 2.80; CI: 1.77–4.45], multiple falls [aOR: 5.27; CI: 2.74–10.12] and fall-injury [aOR: 2.99; CI: 1.55–5.75] among older adults who reported very good SRH. The odds of multiple falls among older adults with multimorbidity were significantly higher in reference to the odds of multiple falls [aOR: 1.21; CI: 1.01–1.44] among older adults with no multimorbidity.

## Discussion

### Prevalence of falls and fall-related injuries in older Indian adults

Old age, inefficient caregiving and support, persisting health adversities, chronic conditions, fragility of organs, abnormal gait, and physical frailty are some of the presumed causes that are attributed to falls and fall-related injuries among the older adults [10, 20, 27–29]. The present study using country-representative survey information of older Indian adults aged 60 years

**Table 2. Prevalence of fall, multiple falls and fall-injury by background characteristics among older adults.**

| Variables | Fall | | Multiple falls | | Fall-injury | |
|---|---|---|---|---|---|---|
| | w% | p-value | w% | p-value | w% | p-value |
| **Frailty status** | | <0.001 | | <0.001 | | <0.001 |
| No | 11.85 | | 5.25 | | 5.29 | |
| Yes | 15.43 | | 7.73 | | 6.68 | |
| **Age (in years)** | | 0.047 | | 0.679 | | <0.001 |
| 60–69 | 12.42 | | 5.76 | | 5.33 | |
| 70–79 | 12.52 | | 5.03 | | 6.08 | |
| 80+ | 14.12 | | 6.68 | | 7.21 | |
| **Sex** | | <0.001 | | <0.001 | | <0.001 |
| Male | 11.69 | | 5.28 | | 4.91 | |
| Female | 13.52 | | 5.98 | | 6.56 | |
| **Marital status** | | <0.001 | | 0.001 | | <0.001 |
| Currently in union | 12.04 | | 5.37 | | 5.21 | |
| Widowed | 13.87 | | 6.22 | | 6.87 | |
| Others | 9.96 | | 4.33 | | 3.58 | |
| **Living arrangement** | | <0.001 | | 0.001 | | <0.001 |
| Alone | 14.78 | | 5.76 | | 8.31 | |
| With spouse | 12.62 | | 5.82 | | 5.34 | |
| With spouse and children | 11.62 | | 5.06 | | 5.08 | |
| Others | 13.56 | | 6.26 | | 6.44 | |
| **Educational status** | | <0.001 | | <0.001 | | <0.001 |
| None | 13.16 | | 5.91 | | 6.47 | |
| Primary | 13.90 | | 6.40 | | 5.80 | |
| Secondary/higher | 10.63 | | 4.55 | | 4.18 | |
| **Work status** | | <0.001 | | 0.001 | | <0.001 |
| Never worked | 11.30 | | 4.84 | | 5.31 | |
| Not working | 13.43 | | 5.80 | | 6.56 | |
| Working | 13.41 | | 6.31 | | 5.56 | |
| Retired | 10.11 | | 4.88 | | 4.23 | |
| **Self-rated health** | | <0.001 | | <0.001 | | <0.001 |
| Very good | 7.70 | | 2.11 | | 3.84 | |
| Good | 9.02 | | 3.16 | | 4.18 | |
| Fair | 13.5 | | 3.27 | | 5.95 | |
| Poor | 16.95 | | 9.06 | | 6.83 | |
| Very poor | 19.06 | | 11.72 | | 10.28 | |
| **Multi-morbid** | | <0.001 | | <0.001 | | <0.001 |
| No | 12.31 | | 5.50 | | 5.40 | |
| Yes | 15.00 | | 8.00 | | 6.61 | |
| **MPCE quintile** | | 0.031 | | 0.031 | | 0.741 |
| Poorest | 11.5 | | 4.44 | | 5.38 | |
| Poorer | 13.64 | | 5.88 | | 6.17 | |
| Middle | 11.86 | | 5.53 | | 5.59 | |
| Richer | 12.72 | | 5.52 | | 6.00 | |
| Richest | 13.67 | | 7.24 | | 5.62 | |
| **Caste** | | 0.001 | | 0.160 | | <0.001 |
| SC/ST | 13.26 | | 6.34 | | 5.61 | |
| OBC | 11.71 | | 4.89 | | 5.83 | |

(*Continued*)

**Table 2.** (Continued)

| Variables | Fall | | Multiple falls | | Fall-injury | |
|---|---|---|---|---|---|---|
| | w% | p-value | w% | p-value | w% | p-value |
| Others | 13.5 | | 6.18 | | 5.78 | |
| **Religion** | | <0.001 | | <0.001 | | 0.217 |
| Hindu | 12.59 | | 5.66 | | 5.70 | |
| Muslim | 12.28 | | 5.65 | | 5.42 | |
| Others | 13.67 | | 5.41 | | 7.01 | |
| **Place of residence** | | <0.001 | | <0.001 | | <0.001 |
| Urban | 10.35 | | 4.62 | | 4.77 | |
| Rural | 13.57 | | 6.06 | | 6.16 | |
| **Region** | | <0.001 | | <0.001 | | <0.001 |
| North | 10.25 | | 4.65 | | 4.62 | |
| Central | 13.79 | | 6.01 | | 7.09 | |
| East | 15.47 | | 7.12 | | 6.50 | |
| Northeast | 9.89 | | 4.57 | | 4.15 | |
| South | 8.44 | | 2.80 | | 4.43 | |
| West | 15.16 | | 7.83 | | 5.98 | |
| **Total** | 12.63 | | 5.64 | | 5.76 | |

Notes: %: Percentage; w%: weighted percentage prevalence to account for survey design and provide national population estimates; MPCE: Monthly per capita consumption expenditure

and above, shows the associations of multiple socioeconomic and demographic measures with fall, multiple falls, and fall related injuries. We found that 12.63% of older adults have fallen in the last 2 years as opposed to 5.64% who have reported having multiple falls and 5.76% with fall-related injuries in comparison to other studies in developing countries [30–32] elucidating the occurrence of fall among the ageing population. As opposed to the current Indian literature [20, 33–35] that reported a higher incidence of falls among older adults in several regions of the country, our research showed that the prevalence of fall, multiple falls, and fall related injuries were relatively lower among individuals aged 60 and above in the Indian context.

Further, falling among older adults can have direct implications such as immediate hospitalizations, disabilities, and premature death [36], suggesting an urgent need for effective health interventions to prevent falls within the ageing population. According to previous studies, incidence of fall increases the likelihood for multiple falls owing to loss of confidence, bruises, fractures, and cognitive impairment [37–39], however, in our study, we observed that there is a rather lower reporting of multiple falls and fall-related injuries reported by individuals. This could be presumably due to underreporting of falls and fall related injuries by older adults due to their limited ability to recall as has been indicated in prior studies [40, 41] which often leads to a false positive history of falls making it difficult for health professionals to predict the count of future falls of the ageing population. Thus, it is important for health care providers and practitioners to recognize the gap between reporting and underreporting of fall injuries while framing policies targeted at fall prevention and control.

## Prevalence and associations of physical frailty and other predictor variables with fall outcomes

The notion of physical frailty has emerged as a significant area of interest among health scholars and practitioners. A recent study undertaken among the community-dwelling rural older

**Table 3. Multivariable logistic regression estimates of fall, fall-injury and multiple falls by socioeconomic and health characteristics among older adults.**

| Variables | | Fall | Multiple falls | Fall-injury |
|---|---|---|---|
| | | aOR (95% CI) | aOR (95% CI) | aOR (95% CI) |
| Frailty status | No | Ref. | Ref. | Ref. |
| | Yes | 1.24*** (1.09–1.41) | 1.24* (1.05–1.48) | 1.21* (1.01–1.45) |
| Age (in years) | 60–69 | Ref. | Ref. | Ref. |
| | 70–79 | 0.92 (0.80–1.04) | 0.98 (0.81–1.19) | 0.79* (0.65–0.96) |
| | 80+ | 0.96 (0.78–1.18) | 1.01 (0.78–1.30) | 0.99 (0.72–1.37) |
| Sex | Male | Ref. | Ref. | Ref. |
| | Female | 1.23** (1.07–1.41) | 1.37** (1.13–1.67) | 1.17 (0.96–1.42) |
| Education | No education | Ref. | Ref. | Ref. |
| | Primary | 0.93 (0.78–1.10) | 0.84 (0.67–1.06) | 0.79 (0.62–1.00) |
| | Secondary/higher | 0.76* (0.59–0.99) | 0.56** (0.38–0.83) | 0.71 (0.49–1.02) |
| Marital status | Currently in union | Ref. | Ref. | Ref. |
| | Widowed | 0.56* (0.32–0.97) | 0.56 (0.26–1.23) | 0.53 (0.25–1.11) |
| | Others | 0.45* (0.23–0.87) | 0.32* (0.13–0.83) | 0.42 (0.16–1.08) |
| Living arrangement | Alone | Ref. | Ref. | Ref. |
| | With spouse | 0.51* (0.28–0.92) | 0.42* (0.19–0.95) | 0.54 (0.24–1.18) |
| | With spouse and children | 0.47* (0.26–0.85) | 0.41* (0.18–0.92) | 0.48 (0.22–1.04) |
| | Others | 0.99 (0.78–1.26) | 0.83 (0.61–1.13) | 1.15 (0.83–1.61) |
| Work status | Never worked | Ref. | Ref. | Ref. |
| | Not working | 1.30** (1.11–1.53) | 1.41** (1.13–1.75) | 1.30* (1.04–1.63) |
| | Working | 1.55*** (1.31–1.85) | 1.62*** (1.28–2.04) | 1.67*** (1.30–2.13) |
| | Retired | 1.19 (0.90–1.59) | 1.37 (0.94–1.99) | 1.27 (0.83–1.93) |
| Self-rated health | Very good | Ref. | Ref. | Ref. |
| | Good | 1.12 (0.79–1.59) | 1.47 (0.82–2.64) | 1.03 (0.65–1.64) |
| | Fair | 1.66** (1.18–2.35) | 2.75*** (1.55–4.87) | 1.46 (0.93–2.29) |
| | Poor | 2.25*** (1.57–3.22) | 3.84*** (2.13–6.92) | 1.81* (1.11–2.94) |
| | Very poor | 2.80*** (1.77–4.45) | 5.27*** (2.74–10.12) | 2.99** (1.55–5.75) |
| Multi-morbid | No | Ref. | Ref. | Ref. |
| | Yes | 1.11 (0.97–1.27) | 1.21* (1.01–1.44) | 1.18 (0.98–1.43) |
| Wealth quintile | Poorest | Ref. | Ref. | Ref. |
| | Poorer | 1.27** (1.07–1.51) | 1.24 (0.97–1.58) | 1.41** (1.09–1.82) |
| | Middle | 1.11 (0.92–1.33) | 1.10 (0.85–1.42) | 1.38* (1.06–1.80) |
| | Richer | 1.26* (1.05–1.51) | 1.28* (1.00–1.64) | 1.48** (1.15–1.90) |
| | Richest | 1.44*** (1.19–1.74) | 1.23 (0.95–1.59) | 2.15*** (1.65–2.80) |
| Religion | Hindu | Ref. | Ref. | Ref. |
| | Muslim | 0.88 (0.73–1.06) | 0.82 (0.64–1.05) | 0.91 (0.70–1.18) |
| | Others | 1.17 (0.95–1.43) | 1.40* (1.08–1.83) | 0.95 (0.70–1.30) |
| Caste | SC/ST | Ref. | Ref. | Ref. |
| | OBC | 1.00 (0.86–1.16) | 1.24 (0.99–1.54) | 0.87 (0.70–1.09) |
| | Others | 1.16 (0.98–1.37) | 1.30* (1.01–1.67) | 0.98 (0.78–1.25) |
| Place of residence | Urban | Ref. | Ref. | Ref. |
| | Rural | 1.16* (1.01–1.33) | 1.05 (0.86–1.29) | 1.06 (0.86–1.29) |

(*Continued*)

**Table 3.** (Continued)

| Variables | | Fall | Multiple falls | Fall-injury |
|---|---|---|---|---|
| | | aOR (95% CI) | aOR (95% CI) | aOR (95% CI) |
| Region | North | Ref. | Ref. | Ref. |
| | Central | 1.48*** (1.23–1.78) | 1.65*** (1.28–2.14) | 1.40* (1.06–1.83) |
| | East | 1.62*** (1.37–1.91) | 1.44** (1.14–1.82) | 1.64*** (1.29–2.07) |
| | Northeast | 1.01 (0.80–1.27) | 0.96 (0.69–1.34) | 1.04 (0.74–1.46) |
| | West | 0.74** (0.61–0.90) | 0.77 (0.58–1.02) | 0.55*** (0.42–0.73) |
| | South | 1.66*** (1.37–2.00) | 1.34* (1.01–1.79) | 1.81*** (1.38–2.37) |
| Constant | | 0.07*** (0.03–0.14) | 0.02*** (0.01–0.05) | 0.03*** (0.01–0.09) |

Notes:

*if p-value <0.05,

** if p-value <0.005,

*** if p-value <0.001;

aOR: OR adjusted for all the covariates; MPCE: Monthly per capita consumption expenditure

people in the region of South India estimated that 28% of older adults were physically frail, 59% were found to have accumulation of deficits that involved medically diagnosed conditions, BMI, grip strength, general health etc., and 63% were frail under the multi-domain definition of frailty that comprised a physical, psychological, and social component [42]. In a cross-country study undertaken by Biritwum et al. (2016), it was found that India has the prevalence of frailty among its ageing population as opposed to China, Ghana, Mexico, Russia and South Africa [43]. Further, a systematic review analysis of physical frailty among community-dwelling adults undertaken by Nguyen et al. (2015) [44] found that the prevalence of frailty within developing countries was much higher than developed nations. Another study undertaken by Khandelwal et al. (2012) showed that a proportion of 32.3% of older adults above the age of 60 years were physically frail [45]. Likewise, among other developing countries, Brazil has one of the highest proportions of physically frail older adults which is projected to be around 55% [46]. The higher prevalence of physical frailty in our study could be possibly due to higher food insecurity, poor nutritional health, disability, and a higher prevalence of physical and agricultural toil during individual lifetimes in developing countries including India [44], which needs to be further investigated in future studies.

In accordance with earlier work [10, 29, 42, 47], we also found that individuals who are physically frail have a higher odds of falls and related injuries. Several longitudinal prospective and retrospective cohort studies using different measurements of frailty reported that frailty is positively associated with falls [48–50]. The meta-analysis results of a study revealed that compared to robust older adults, older adults entering pre-frail and frail stages are more likely to experience recurrent falls [51]. A review article suggests that the processing functions of frailer older adults might be on the brink of failure and any small stressor may precipitate falls and injuries in these individuals [52]. Furthermore, the specific components of physical frailty such as weight loss and impaired balance may have differential impact on fall outcomes [53], which can be explored in future research.

The findings from this study are consistent with prior research [54, 55], suggesting the higher rate of falls by increasing age. Another important finding of this study points to the strong statistically significant association between richest individuals and fall as well as fall related injuries as compared to other individuals in lower wealth quintiles. This varied from previous findings within India and other countries that showed the higher prevalence of falls

and injuries among poor socioeconomic groups [27, 56–58]. This could be attributed to the fact that wealthier individuals have a higher likelihood of living within nuclear family setups and young adults are likely to move away (abroad) from such families, thus leaving their older parents behind. Thus, it could also be translated directly to the caregiving crisis and insufficient support from immediate family members increasing the likelihood of associated falls among poor older adults. Further, living alone, loneliness, depression, anxiety, and isolation are other potential factors that may contribute to falls and injuries within the ageing population [59, 60]. Thus, older adults who are living alone in the absence of any caregiving facilities should be institutionalized and healthcare professionals should factor in the loneliness and social isolation components among higher socioeconomic groups while framing fall risk assessment tools and strategies.

Our findings also showed that older females had a far more likelihood of falls and multiple falls than males, as evidenced by prior studies [61, 62]. This could be possibly due to poor grip strength, lower average lean body mass, and higher threats of sarcopenia among women as compared to men [63]. Further, individual work status was an important moderating factor as we observed statistically significant association between working individuals with falls, multiple falls, and fall related injuries. This could be attributed to possibilities of agricultural work among older farmers who spend a majority of their time in the fields with long working hours and higher intensity of physical activities. Unemployed older individuals, on the other hand, have lesser exposure to intensive physical labor thereby enabling them to spend time within the comfort of their homes and thus, minimizing the risk of falls and its impacts. We also found that the prevalence of falls and multiple falls was higher among those with higher education, contrary to previous studies within the Indian context [64, 65], however, supported by another study [66] that found the increased likelihood of falls with higher education. Although it may partially be explained by the increased awareness and reporting among educated people, the possible association between education and any or multiple falls have not been identified previously and there still remains gap for possible exploration.

Further, our findings showed that risk of falls, multiple falls and fall related injuries was greater among older adults with poor health conditions. In other words, individuals who reported poor to very poor SRH and those who were multi-morbid had higher rate of falls, multiple falls and fall-related injuries, as evidenced from previous findings [67, 68]. This could be attributed to the balance confidence of individuals, including their recent repeated hospital visits [69] and functional impairments due to poor SRH and morbidity [70]. However, given the subjective nature of SRH which taps into the psychology of an individual, other relevant factors could also be investigated when examining the associations between SRH and fall related outcomes among older adults. A recent study revealed that bone and joint diseases are positively associated with falls among older Indian adults [65]. Thus, other specific chronic conditions associated with fall outcomes can also be explored in future studies. Additionally, it was interesting to note that fall outcomes varied with regard to different regional contexts, which has not been widely explored within the current literature and thus, calls for special attention on identifying the causes for the increasing fall injuries in specific regions.

## Strengths and limitations of the study

The current study contributes to the wide range of literature that explores the association between physical frailty and fall related outcomes within developing countries and reported a higher incidence of fall rates among older adults [20, 34]. Further, the large sample size collected from older individuals within different regional contexts and socio-economic backgrounds points to the higher range of generalizability and representativeness of the data. In

addition to these strengths, there are certain limitations of the study which should also be acknowledged. First and important is the cross-sectional design of the study which prohibits the drawing of any causal inferences in the observed associations. This suggests that as documented in multiple previous studies, fall outcomes may result in increased chances of physical frailty among older adults. Second, the lack of a universally accepted method of reporting falls can intuitively obstruct older adults from giving correct information. Third, since the prevalence of falls is very likely to be underreported, it can possibly hamper the reliability of our results, thereby reducing its strengths and significance throughout our study. Finally, the subjective nature of some of the socio-demographic variables (wealth status) as well as SRH might be inadequate in the identification of respective sample population and also open ways to multiple interpretations of the outcomes and associations observed.

## Conclusion

Older individuals with physical frailty were found to be at increased risk of fall, multiple falls and fall-related injury in the current study. The findings of our study also have important clinical implications in the measures undertaken to reduce falls and enable future healthcare practitioners and policymakers to factor in the key determinant of physical frailty. Older adults who are employed, have a higher wealth status or report a lower SRH are also potentially at a higher risk of fall and related injuries, and thus, those individuals should be the target groups and likely deserve special attention from health practitioners and caregivers.

## Acknowledgments

The authors are thankful to the International Institute for Population Sciences, Mumbai for providing the LASI data for undertaking this study.

## Author Contributions

**Conceptualization:** Shriya Thakkar, Muhammad T., Shobhit Srivastava.

**Data curation:** Muhammad T.

**Formal analysis:** Muhammad T.

**Investigation:** Muhammad T.

**Methodology:** Muhammad T.

**Software:** Muhammad T.

**Supervision:** Shriya Thakkar, Muhammad T., Shobhit Srivastava.

**Validation:** Shriya Thakkar, Muhammad T., Shobhit Srivastava.

**Writing – original draft:** Shriya Thakkar, Muhammad T., Shobhit Srivastava.

**Writing – review & editing:** Shriya Thakkar, Muhammad T., Shobhit Srivastava.

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
