## [Decision Letter · Decision Letter 0]

4 Jul 2022

PONE-D-22-10504Cross-sectional associations of frailty with falls, multiple falls and fall-injury among older Indian adults: findings from LASI, 2018PLOS ONE

Dear Dr. T.,

Thank you for submitting your manuscript to PLOS ONE. After careful consideration, we feel that it has merit but does not fully meet PLOS ONE’s publication criteria as it currently stands. Therefore, we invite you to submit a revised version of the manuscript that addresses the points raised during the review process.

The manuscript has received encouraging reviews. Please address the issues of the reviewers and we will reconsider the manuscript. 

We look forward to receiving your revised manuscript.

Kind regards,

David G. Greenhalgh, MD

Academic Editor

PLOS ONE

Journal Requirements:

Additional Editor Comments:

Editor: There are positive reviews for this manuscript. Please address the reviewer's critiques.

Reviewers' comments:

Reviewer's Responses to Questions

**Comments to the Author**

1. Is the manuscript technically sound, and do the data support the conclusions?

Reviewer #1: Yes

Reviewer #2: Yes

2. Has the statistical analysis been performed appropriately and rigorously? 

Reviewer #1: I Don't Know

Reviewer #2: Yes

3. Have the authors made all data underlying the findings in their manuscript fully available?

Reviewer #1: No

Reviewer #2: Yes

4. Is the manuscript presented in an intelligible fashion and written in standard English?

Reviewer #1: Yes

Reviewer #2: Yes

5. Review Comments to the Author

Reviewer #1: Thank you for the opportunity to review this manuscript. It is well written and on an important topic. My suggestions for revisions include:

Abstract:

1.) Need to define the term SRH in the abstract

Background

2.) Please state your hypothesis in the background

3.) In the conceptual framework you need to define the acronym MPCE

4.) Please give a more detailed account of the tests used in your statistical analysis

5.) SRH needs to be defined in the text

Reviewer #2: This is a well conducted study in a large population. This study highlights that frailty is not only a clinical and social problem for aging western developed countries but likely a major problem for other eastern large countries as well. The authors paper is well written and data analysis and conclusions are sound.

6. PLOS authors have the option to publish the peer review history of their article (what does this mean?). If published, this will include your full peer review and any attached files.

Reviewer #1: No

Reviewer #2: No

---

## [Author Response · Author response to Decision Letter 0]

13 Jul 2022

Title: Cross-sectional associations of frailty with fall, multiple falls and fall-injury among older Indian adults: findings from LASI, 2018

Editor comments

The manuscript has received encouraging reviews. Please address the issues of the reviewers and we will reconsider the manuscript. 

Response: Dear Editor, the authors are thankful to you and both the reviewers for pointing out the merits of the study and recommending the paper to be published in the journal after incorporating the comments. 

Review Comments to the Author

Reviewer #1: 

Thank you for the opportunity to review this manuscript. It is well written and on an important topic. My suggestions for revisions include:

Abstract:

1.) Need to define the term SRH in the abstract

Response: The SRH is defined now at its first use.

Background

2.) Please state your hypothesis in the background

Response: The hypotheses of the study are mentioned now in the background.

3.) In the conceptual framework you need to define the acronym MPCE

Response: The MPCE is defined in the revised framework.

4.) Please give a more detailed account of the tests used in your statistical analysis

Response: More details are provided for the tests used in the statistical analyses.

5.) SRH needs to be defined in the text

Response: SRH is defined at first use now.

Reviewer #2: 

This is a well conducted study in a large population. This study highlights that frailty is not only a clinical and social problem for aging western developed countries but likely a major problem for other eastern large countries as well. The authors paper is well written and data analysis and conclusions are sound.

Response: Dear reviewer, many thanks for pointing out the merits of the study and the recommendation.

---

## [Decision Letter · Decision Letter 1]

25 Jul 2022

Cross-sectional associations of physical frailty with fall, multiple falls and fall-injury among older Indian adults: findings from LASI, 2018

PONE-D-22-10504R1

Dear Dr. T.,

We’re pleased to inform you that your manuscript has been judged scientifically suitable for publication and will be formally accepted for publication once it meets all outstanding technical requirements.

Kind regards,

David G. Greenhalgh, MD

Academic Editor

PLOS ONE

Additional Editor Comments (optional):

Accept

Reviewers' comments:

Reviewer's Responses to Questions

**Comments to the Author**

1. If the authors have adequately addressed your comments raised in a previous round of review and you feel that this manuscript is now acceptable for publication, you may indicate that here to bypass the “Comments to the Author” section, enter your conflict of interest statement in the “Confidential to Editor” section, and submit your "Accept" recommendation.

Reviewer #1: All comments have been addressed

Reviewer #2: All comments have been addressed

2. Is the manuscript technically sound, and do the data support the conclusions?

Reviewer #1: Yes

Reviewer #2: Yes

3. Has the statistical analysis been performed appropriately and rigorously? 

Reviewer #1: Yes

Reviewer #2: Yes

4. Have the authors made all data underlying the findings in their manuscript fully available?

Reviewer #1: Yes

Reviewer #2: Yes

5. Is the manuscript presented in an intelligible fashion and written in standard English?

Reviewer #1: Yes

Reviewer #2: Yes

6. Review Comments to the Author

Reviewer #1: (No Response)

Reviewer #2: Well written article, the objectives, data collection and statistical analysis are appropriate for this study. The conclusions are supported by the data and analysis

7. PLOS authors have the option to publish the peer review history of their article (what does this mean?). If published, this will include your full peer review and any attached files.

Reviewer #1: **Yes: **Kathleen Skipton Romanowski

Reviewer #2: No

---

## [Editor Report · Acceptance letter]

1 Aug 2022

PONE-D-22-10504R1 

Cross-sectional associations of physical frailty with fall, multiple falls and fall-injury among older Indian adults: findings from LASI, 2018 

Dear Dr. T.:

I'm pleased to inform you that your manuscript has been deemed suitable for publication in PLOS ONE. Congratulations! Your manuscript is now with our production department. 

Kind regards, 

on behalf of

Dr. David G. Greenhalgh 

Academic Editor

PLOS ONE